# Reliability and Validity Estimate of the Pro-Inflammatory/Anti-Inflammatory Food Intake Score in South American Pediatric Population: SAYCARE Study

**DOI:** 10.3390/ijerph20021038

**Published:** 2023-01-06

**Authors:** Lívia Gabriele Azevedo-Garcia, Francisco Leonardo Torres-Leal, Juan Carlos Aristizabal, Gabriela Berg, Heráclito B. Carvalho, Augusto César Ferreira De Moraes

**Affiliations:** 1YCARE (Youth/Child and Cardiovascular Risk and Environmental) Research Group, Faculdade de Medicina, Universidade de Sao Paulo, Sao Paulo 01246-903, Brazil; 2Metabolic Diseases, Exercise and Nutrition (DOMEN) Research Group, Federal University of Piauí, Teresina 64049-550, Brazil; 3Grupo de Investigación en Fisiología y Bioquímica (PHYSIS), Universidad de Antioquia, Medellín 050010, Colombia; 4Departamento de Bioquímica Clínica, Facultad de Farmacia y Bioquímica, Universidad de Buenos Aires, Cátedra de Bioquímica Clínica I, Junín 956, Buenos Aires C1113AAD, Argentina; 5Facultad de Farmacia y Bioquímica, Instituto de Fisiopatología y Bioquímica Clínica (INFIBIOC), Universidad de Buenos Aires, Buenos Aires C1120AAF, Argentina; 6Faculty of Pharmacy and Biochemistry, University of Buenos Aires, Junín 956, Buenos Aires C1113AAD, Argentina; 7The University of Texas Health Science Center at Houston, School of Public Health Austin Campus, Michael & Susan Dell Center for Healthy Living, Austin, TX 78701, USA; 8Graduate Program in Public Health, Graduate Program in Epidemiology, School of Public Health, University of Sao Paulo, Sao Paulo 01246-904, Brazil

**Keywords:** validation study, dietary inflammatory index, food frequency questionnaire, inflammatory biomarker, inflammation, pediatric

## Abstract

Chronic low-grade inflammation may be associated with the development of chronic non-communicable diseases in young populations, often lasting to adulthood. Studies show that the diet is related to chronic inflammation. The Pro-inflammatory/Anti-inflammatory Food Intake Score (PAIFIS) is an indicator that measures the inflammatory potential of the diet, with the help of validated tools that assess food consumption. The validation of tools that assess inflammatory dietary patterns in young populations to produce valid and reliable results is essential to guide disease prevention strategies for adulthood. Methods: This study aimed to estimate the Pro-inflammatory/Anti-inflammatory Food Intake Score (PAIFIS) in children and adolescents in South America and to test its reliability and validity using a food frequency questionnaire (FFQ) and an inflammatory biomarker. This work consists of a validation study in a sample of children and adolescents conducted in South America (SAYCARE Study). The habitual consumption of food contributing to calculating the PAIFIS was obtained through an FFQ and 24 h Dietary Recall (24HDR). Reliability was tested using the FFQ (FFQ1 × FFQ2), using Spearman’s correlation coefficient to estimate the agreement between measurements. The validity of the PAIFIS was tested using 24HDR and the inflammatory biomarker C-reactive protein (CRP) using Spearman’s correlation and multilevel linear regression. Results: For children and adolescents, pro- and anti-inflammatory food groups showed Spearman’s correlation coefficients ranging from 0.31 to 0.66, convergent validity ranging from 0.09 to 0.40, and criterion validity for a reliability range from −0.03 to 0.18. The PAIFIS showed Spearman’s correlation coefficients for reliability ranging from 0.61 to 0.69, convergent validity from 0.16 to 0.23, and criterion validity from −0.03 to 0.24. Conclusion: The PAIFIS showed acceptable reliability, weak convergent validity, and weak criterion validity in children and adolescents.

## 1. Introduction

Chronic inflammation has a strong association with lifestyle behaviors [1]. Among these behaviors, diet is directly associated with inflammatory markers [2,3]. Multicenter epidemiological studies suggest that a diet rich in raw vegetables, olive oil, fish, whole grains, and fruits is associated with low levels of inflammation and that a diet rich in red meat and high in saturated fat, refined carbohydrates, dairy products, and sugary drinks is associated with a state of chronic inflammation with high levels of some inflammatory biomarkers, such as IL-1, IL-4, IL-6, IL-10, high-sensitivity C-reactive protein (hs-CRP), and TNF-alpha [4,5,6]. The literature suggests that the most commonly used inflammatory biomarker is hs-CRP [7,8].

Cavicchia, in 2009, developed the Dietary Inflammatory Index (DII), which was later updated in 2014 by Shivappa [9,10]. It is a tool with the potential to estimate the effect of diet on circulatory inflammation. This index has shown an association with serum levels of hs-CRP, IL-6, and TNF-α [11,12,13]. Many derivations of the index created in 2009 have been developed, such as the E-DII (Energy-density Dietary Inflammatory Index), which is derived using procedures like those for the DII but is an index adjusted for energy, the C-DII (Children’s Dietary Inflammatory Index), which is an index validated for North American children, and similar indices that measure potential dietary inflammation in different populations [14].

Most studies published using any type of index that measures the degree of inflammation in the diet were predominantly conducted with adults or the elderly, with no study focusing on children and adolescents in South America [8,15,16]. Knowing that unhealthy eating habits based on pro-inflammatory foods usually begin in childhood and adolescence and develop throughout life, favoring the emergence of chronic diseases in adulthood, the study of diet in populations of children and adolescents using validated instruments becomes crucial. The associations of dietary inflammatory indices with health outcomes are directly dependent on the method of measuring the diet, for example, 24 h Food Recall (24HDR), a food frequency questionnaire (FFQ), and a food diary [17,18,19]. Additionally, epidemiological studies in the population of children and adolescents that estimate the validity and reliability of subjective methods of assessing food intake compared to inflammatory biomarkers are scarce [8].

There is a lack of literature about validated methods of assessing food consumption that measure the inflammatory score of the diet in children and adolescents in Latin America. It is crucial to validate tools that assess inflammatory food patterns in young populations to obtain valid and reliable results for the prevention of chronic diseases in adulthood. The aim of this study was to estimate the Pro-inflammatory/Anti-inflammatory Food Intake Score (PAIFIS) in South American children and adolescents using two subjective methods of dietary intake assessment and to test their reliability and validity using an FFQ and an inflammatory biomarker.

## 2. Methods

### 2.1. Study Design

The SAYCARE Study (“South America Youth/child Cardiovascular and Environment Study”) is an observational, epidemiological, cross-sectional feasibility study that aimed to develop valid and reliable instruments to obtain information about socio-environmental factors, the family environment, food intake, physical activity and inactivity, sedentary behavior, body composition, sleep, oral health, and cardiovascular health biomarkers. The study was carried out in a convenience sample of children and adolescents aged 3 to 18 who attend public and private schools in seven cities of South America: Buenos Aires (Argentina), Lima (Peru), Medellín (Colombia), Montevideo (Uruguay), Santiago (Chile), and São Paulo and Teresina (Brazil). A detailed description of the SAYCARE Study has been published elsewhere [20].

### 2.2. Quality Control of SACYARE Study

This multicenter feasibility study is premised on ensuring that data collection is carried out in a standardized manner, achieving a good representation of reality and allowing the comparison of data from the different cities involved in the study. The data management system was created and led by a general coordinator of the study at the Medical School of the University of São Paulo. Reports from all centers were sent to São Paulo to verify the consistency of the data. For exclusively nutrition data, questionnaire data (FFQ and R24HR) were entered into the software and tabulated by two independently trained nutritionists. If there were any inconsistencies, a data manager checked and corrected the error [20].

In addition, to ensure the quality of the data, standardized procedures were applied, guaranteeing the accuracy and consistency of the results obtained during the study, such as manuals or standardized measurements, registration protocols, training of the entire team, and the supervision of researchers through audit protocols. Inconsistencies were quickly identified and corrected with this procedure, and diagnoses and the resolution of difficulties were anticipated during data collection and processing. A detailed description of the SAYCARE Study’s quality control has been published elsewhere [20].

### 2.3. Study Population and Data Collection for This Study

This study is nested in the SAYCARE Study, using a sub-sample of participants equally distributed by sex and type of school (public and private).

The following exclusion and inclusion criteria were adopted for the sub-sample

Exclusion criteria: inability to answer the questionnaires; lack of the participant’s consent to collect blood samples; and participants with physiologically implausible energy intake for this age group estimated by the FFQ. Inclusion criteria: children and adolescents (3 to 18 years) who had blood collection performed and the inflammatory biomarker assessed, with data from at least two 24HDRs and FFQs and complete information regarding sex, age, weight, and height.

To calculate the sample to analyze the reliability and convergent validity, we used the following parameters: α = 0.05, β = 0.20, and a correlation coefficient of 0.70 [21]; for criterion validity, the parameters were α = 0.05, β = 0.20, and a correlation coefficient of 0.45. We estimated a necessary sample of 130 subjects for reliability and convergent validity and 30 individuals for criterion validity from these parameters. Foreseeing possible losses, the sample collected was 25% larger. These sample sizes were considered adequate according to validation studies of food intake tools and studies using biomarkers [22,23,24].

SAYCARE questionnaires were answered by the parents/caregivers of the children (3–10 years), whereas the adolescents (11–18 years) responded for themselves.

Participants responded to the FFQ twice and at performed least two 24HDRs. Parents or adolescents received verbal and written instructions on how to complete the questionnaires. The instructions included standardized examples of common measures used (cups, spoons, etc.) and a colored photo booklet of foods, which contained photos of commonly consumed foods, including country-specific foods, and their standard portion sizes to facilitate accurate records [25]. Total energy intake was calculated from FFQ data using the United States Department of Agriculture (USDA) food composition database and a SAYCARE country-specific food composition database when local foods were not found in the USDA database.

As highlighted by Carvalho et al. [20], we did not find differences in the main variables that we evaluated (sex, age, and mother’s education in both age groups); however, we found that children and adolescents from public schools had more losses in the second questionnaire.

An important result is that the highest proportions of losses/refusal were from individuals with lower socioeconomic status (SES), indicating that future studies in South America should focus on this SES class. This difficulty is attributed to the fact that because the questionnaire is extensive, the parents of the children and adolescents may have lost the motivation to respond to the same questionnaire twice in a short time. However, our n was within the estimated range.

### 2.4. Food Frequency Questionnaire (FFQ)

The SAYCARE FFQ is a semi-quantitative questionnaire that was developed and validated for children and adolescents in South America, as described elsewhere [25]. This FFQ refers to the usual consumption in the last three months, having a basic core of 47 foods that were common to the centers of the SAYCARE Study. Typical foods from each center were included in their respective FFQ, resulting in 63 items for Buenos Aires, 61 items for Lima, 63 items for Medellín, 59 items for Montevideo, 57 items for Santiago, 67 items for São Paulo, and 69 for Teresina. All research subjects answered the FFQ twice (FFQ1 and FFQ2) with an interval of two weeks to assess the reliability [25].

The difference between the FFQ of children and adolescents is that the alcohol item was included in the adolescent’s questionnaire; additionally, those who answered the children’s questionnaire were parents or guardians, while those already in adolescence answered for themselves. A detailed description of the SAYCARE FFQ is published elsewhere [25].

In order to manage, organize, and facilitate the collection of data, the data collected in FFQ1 and FFQ2 were entered on a secure web platform [20].

The FFQ provided data on portion sizes and daily frequency, allowing the estimation of the daily intake of foods in grams or milliliters for each daily portion, multiplying the frequency of consumption by the portion size. Finally, foods were categorized into pro-inflammatory (sugar-sweetened beverages, processed meats (hamburger, sausage, ham, turkey breast, and bologna/cured meat), red meat, snacks (foods high in fat and salt: snacks in general, chips, pizza, and frozen or ready-to-eat foods), and candies) and anti-inflammatory (fruits and vegetables; Appendix A).

For example, in the case of fruits and vegetables (anti-inflammatory profile), fruits and vegetables were queried separately in different groups, such as fresh fruit, fresh fruit juice, raw vegetables, boiled vegetables, vegetable soup, and tomato sauce. These groups were defined in nine frequencies of intake, as follows: (1) never/less than once a month; (2) 1–3 times per month; (3) once a week; (4) 2–4 times per week; (5) 5–6 times per week; (6) once a day; (7) 2–3 times per day; (8) 4–5 times per day; and (9) 6 or more times per day [25].

If the parent/guardian answered that his/her child usually eats one portion of fruit 2–3 times per day and one portion of vegetables per day, the total fruit and vegetable daily intake was calculated as [fruit: 80 (medium portion) × 2.5 (2–3 times per day) = 200 g/day] + [vegetable: 50 (medium portion) × 1 (once a day) = 50 g/day]. Therefore, the total daily fruit and vegetable (∑AI = anti-inflammatory profile) intake was (200 + 50) 250 g per day.

### 2.5. Twenty-Four-Hour Dietary Recall (24HDR)

24HDR is an open questionnaire about the foods consumed the previous day, which must be described quantitatively and qualitatively using common measures such as cups, spoons, etc., along with the time and meals. Parents of children and adolescents responded to the first and second 24HDRs at school with a trained nutritionist, and the third 24HDR was answered at home along with supporting materials, with previously given instructions (2 non-consecutive weekdays—by trained nutritionists; 1 weekend day—by parents/guardians/adolescent) [21]. Except for Buenos Aires and Lima, caregivers answered the three 24HDRs at home (except in special situations, such as non-literate caregivers who needed help to respond to the questionnaires, and they completed it at the school with the help of a trained dietitian).

The data obtained from the 24HDR questionnaire were input into an Ibero-American platform for food consumption research, which assesses nutrient and energy intake. The food composition databases included were the Spanish Food Composition Database (Bedca), the National Nutrient Database for Standard Reference Release (USDA), and the Ibero-American Food Database. In addition, some typical Brazilian foods were included in this software, considering the Brazilian Table of Food Composition (TACO). Finally, the data from 24HDR were input into the software and tabulated in an Excel spreadsheet, and they were grouped so as to converge with the corresponding items of the FFQ. Data obtained from at least two 24HDRs, in grams (or milliliters, for drinks), were added together, and the average was calculated to provide the average daily intake. Finally, the food items of 24HDR were grouped into pro- and anti-inflammatory foods, as previously described, to allow a direct comparison between FFQ and 24HDR (validity analysis).

### 2.6. Anthropometry

The measurements of body composition were assessed by previously trained professionals using a standardized methodology [20].

These assessments were carried out in a private room at each school, with the children barefoot and wearing lightweight clothing. The analyzed variables, according to the anthropometric standardization reference manual of the World Health Organization [26], were weight measured with a 0.1 kg precision digital scale, height measured with a wall stadiometer with a millimeter scale, and waist circumference measured with flexible tape (in centimeters) [27].

The measurements were performed twice, and a third measurement was made only if the error was greater than 5% between the first two measurements. Weight and height were used to calculate the body mass index (BMI), obtained by dividing the weight (kg) by the squared height (meters) and classified according to the criteria defined by Cole et al., 2007 [28].

### 2.7. Inflammatory Biomarker

In this study, we adopted high-sensitivity C-reactive protein (hs-CRP) as the inflammatory biomarker. hs-CRP concentrations were measured in a central laboratory with a high-sensitivity assay using nephelometry (BN2-Nephelometer, Siemens, Deerfield, IL, USA). The lower limit of detection of the assay was 0.02 mg/dL [29].

### 2.8. Pro-Inflammatory/Anti-Inflammatory Food Intake Score (PAIFIS)

For the calculation of the *PAIFIS*, using the subjective assessment tools for food consumption, the sums of foods, in grams, with pro-inflammatory (*∑PI*) and anti-inflammatory (*∑AI*) profiles were calculated, and after that, the sum of pro-inflammatory foods was subtracted by the sum of anti-inflammatory foods for each participant.
PAIFIS=ΣPI−ΣAI

For example:

First: Calculate the daily consumption of the pro- and anti-inflammatory foods listed in Table 1.

Second: Calculate the sum of the daily consumption in grams of all pro-inflammatory foods.
$$\left(\Sigma PI\right)=(red\;meat+processed\;meat+Sugar-Sweetened\;Beverages\;+candies\;+snacks)$$

Third: Calculate the sum of daily consumption in grams of all anti-inflammatory foods.
$$\left(\Sigma AI\right)=\left(Fresh\;fruits+raw\;and\;cooked\;vegetables\right)$$

Fourth: Subtract the sums of daily consumption in grams of all pro-inflammatory and anti-inflammatory foods.

The existence of pro-inflammatory and anti-inflammatory food profiles for children and adolescents is already known in the literature. These profiles were validated together with pro-inflammatory biomarkers, such as hs-CRP, IL-6, and TNF-alpha [4,5,30]. In some systematic reviews, researchers point out that hs-CRP is the most used biomarker in studies of the association between inflammatory dietary patterns or inflammatory scores and this biomarker [7].

Remember that these pro- and anti-inflammatory profiles’ items are food groups; so, for example, in the fruit category, we have a significant number of a variety of fruits (more than 20 types) and fresh fruit juice, and in the case of vegetables (30 types), we have raw and boiled vegetables, vegetable soups, and tomato sauce (a list of all reported fruits and vegetables is available in Appendix A). As in the pro-inflammatory food group, this is also the case for the anti-inflammatory group: for example, the processed meat group is composed of burger, sausage, ham, turkey breast, and bologna. The items that make up these profiles are shown in Figure 1.

### 2.9. Statistical Analysis

Statistical analyses were performed using Stata software version 14.0 (Stata Corporation, College Station, TX, USA). The criterion for statistical significance was set at 5%.

First, a descriptive analysis of the sample in this study was carried out, which includes the calculation of the mean (standard deviation, SD) for continuous variables and the percentage (%) and 95% confidence interval (95% CI) for categorical variables. The normality of the sample was previously verified by the Shapiro–Wilk test.

The PAIFIS and the daily intake of the individual pro-inflammatory and anti-inflammatory foods that compose it were calculated from the FFQ, and their reliability and validity [31] were determined by Spearman’s correlation coefficients. We estimated the strength of the agreement using correlation coefficient cutoff points defined using the following classification: weak = 0.10 to 0.29, moderate = 0.3 to 0.69, and strong = 0.70 to 1.0 [32]. We considered an acceptable correlation for reliability and validity analysis to be coefficients to be greater than 0.30, as recommended in the literature for lifestyle behaviors [32].

The PAIFIS criterion validity, using the FFQ tool to estimate habitual inflammatory food consumption, was tested against an inflammatory biomarker using Spearman’s correlation and multilevel linear regression models for contextual variables (center and type of school) and individual factors (sex, age, and energy). Participants who reported any type of infection were excluded from these analyses. Participants with hs-CRP > 10 mg/dL were also excluded from the analyses, as it is a value suggestive of active inflammation or infection [33].

## 3. Results

This study aimed to obtain a Pro-inflammatory/Anti-inflammatory Food Intake Score (PAIFIS) with the aid of an FFQ and 24HDR in a population of children and adolescents in South America. We thus provide valid methods for application in studies with various epidemiological designs in South America. Table 1 shows the sample distribution by the SAYCARE Study center for each analysis.

In this study, we used a sub-sample of the SAYCARE Study composed of children and adolescents (3 to 18 years old) from São Paulo, Teresina, Buenos Aires, Medellín, Lima, Montevideo, and Santiago, as featured in Table 1. The flowchart that indicates the sampling process according to the type of analysis to be performed is shown in Figure 1; we obtained an initial sample of 371 children and 290 adolescents who answered FFQ1 and had valid data.

Even with the reduction in participants due to losses, the sample was sufficient to conduct reliability and validity analyses. Additionally, for criterion validity, a post hoc analysis was performed, considering a two-tailed error of 0.05 (type I), a β error of 0.20 (type II error), and a correlation coefficient of 0.30, resulting in 8 participants for each center, totaling 32 children and 32 adolescents.

The descriptive analyses for reliability and validity are shown in Table 2, divided by the age group, according to sex, type of school, BMI, waist circumference, and hs-CRP. Regardless of the type of analysis, at least 20% of the children and at least 30% of the adolescents were overweight/obese.

Comparing the median food consumption of the pro-inflammatory profile (ΣPI) obtained by FFQ1, the consumption obtained by children was 440.5 g, and for adolescents, it was 491.3 g; it is possible to perceive a higher consumption of this food profile by adolescents. The same happens when we observe the consumption obtained by FFQ2, where children ΣPI = 441.5 g, and adolescents ΣPI = 501.2 g. When comparing the median food consumption of the pro-inflammatory profile (ΣPI) obtained by R24HR, children consumed 559.0 g and adolescents consumed 740.8 g, so a higher consumption of this profile by adolescents is observed.

When analyzing the median consumption of foods with an anti-inflammatory profile (ΣPA) obtained by FFQ1, children consumed 283.1 g and adolescents consumed 304.4 g, so a higher consumption of these foods by adolescents is noted. The same is observed for FFQ2, where for children, ΣPA = 212.5 g, and for adolescents, ΣPA = 238.1 g. For the same anti-inflammatory profile (ΣPA) but observing the median consumption obtained by R24HR, it is verified that children consumed 215.1 g and adolescents consumed 163.3 g, showing a higher consumption of this food profile in children than in adolescents.

We found moderate reliability in the pro- and anti-inflammatory food groups according to Spearman’s coefficients ρ (ranging from 0.51 to 0.66 in children and 0.31 to 0.59 in adolescents). For the PAIFIS, we had moderate reliability with a coefficient of 0.69 for children and 0.61 for adolescents, as shown in Table 3.

We found weak to moderate convergent validity in the pro- and anti-inflammatory food groups according to Spearman’s coefficients (ranging from 0.40 to 0.20 in children and 0.29 to 0.09 in adolescents). For the PAIFIS, we had weak convergent validity for children (ρ = 0.23) and adolescents (ρ = 0.16), as shown in Table 4.

For criterion validity, in the pro- and anti-inflammatory food groups, Spearman’s coefficients of 0.01 and 0.03, respectively, were found for children, and 0.16 and 0.07, respectively, were found for adolescents. In both age groups, the correlation coefficients were weak. Regarding the PAIFIS for children, a weak correlation between the score and the biomarker (ρ = −0.03) was found, and a weak correlation was found for adolescents (ρ = 0.24), as highlighted in Table 5.

The multilevel regression results showed that the variance between the tools for calculating the PAIFIS, FFQ compared to R24HDR, was 43.3% for children, with a regression coefficient (R) of 0.32. For adolescents, we observed a variance of 15.5% and a regression coefficient (R) of 0.24. On the other hand, when analyzing criterion validity, comparing the PAIFIS with the hs-CRP biomarker, the variance presented for children was 43.9% with a regression coefficient (R) 0.48, whereas for adolescents, the variance was 61.7% and the regression coefficient (R) was 0.33, as highlighted in Table 5.

## 4. Discussion

We tested the reliability and validity of a newly developed FFQ and 24HDR to classify the dietary intake of PAIFIS among South American children and adolescents. The Pro-inflammatory/Anti-inflammatory Food Intake Score proved to be reliable for South American children and adolescents. The convergent validity was weak for children and adolescents, with weak criterion validity for children and adolescents. It is shown to be a reliable tool to estimate the inflammatory diet score from an FFQ in the pediatric population.

### 4.1. Reliability

It was observed that FFQ1 presented higher estimates of the consumption of some groups of inflammatory foods in both age groups. However, this difference is only noticed in children, as evidenced by the PAIFIS, presenting a Δ = 67.1 g between the frequency questionnaires. The Pro-inflammatory/Anti-inflammatory Food Intake Score showed acceptable reliability with Spearman’s coefficients of 0.69 for children and 0.61 for adolescents. As in the SAYCARE FFQ, higher estimates of food consumption in the first FFQ are also evidenced in other studies [21,34,35].

### 4.2. Convergent Validity

R24hrs showed higher estimates of consumption in some groups of inflammatory foods and lower ones in others when compared to FFQ1, reflecting higher values of PAIFIS in children (Δ = 278.0 g) and adolescents (Δ = 315.9 g). The opposite is observed in the literature, with higher estimates of food consumption in FFQ1 compared to R24hrs [36,37,38]. The PAIFIS showed weak convergent validity for children (0.23) and adolescents (0.16).

### 4.3. Measurement Bias

There are studies that evaluate the validity of subjective food consumption tools in children and adolescents but with inconsistent results, which may be due to indirect reporting by parents [39], the eating habits of the studied population [37], or the fact that, although R24hrs is generally used as a gold standard, it is not a perfect measure of dietary intake [34,37,39]. Among children, consumption is usually reported by parents/guardians, resulting in this report often being unreliable [39] due to factors such as the parent’s educational level [40], a desire to influence the report of what was consumed [41], forgetfulness [21], and non-participation in the child’s meal routine, generating inaccurate reports, which can lead to unacceptable validity coefficients (ρ < 0.30) [22,42]. In adolescence, forgetfulness or remembering only some of the foods consumed may occur, in addition to difficulty in estimating portion sizes [43,44].

### 4.4. Validity Criterion

For criterion validity between the PAIFIS and the inflammatory biomarker (CRP), Spearman’s coefficients ranging from −0.03 to 0.24 were found. A weak correlation was observed in children and in adolescents.

When studying relationships between dietary intake and disease development in epidemiology, it is important to account for the dietary assessment method’s ability to correctly categorize participants by dietary intake. Studies to assess the reliability of food frequency questionnaires in a free-living environment will be explored in comparison with instruments that provide objective measures to improve the accuracy of subjective measures that are common in nutritional epidemiology [22]. The SAYCARE FFQ was validated for fruit and vegetable intake [24], as well as for iron intake [45], with very good coefficients (range of 0.35 to 0.85). Consequently, the FFQ should not be used as a quantitative tool to measure the anti-/pro-inflammatory status of pediatric populations.

There are few studies in the literature that validate indices calculated using subjective tools with inflammatory biomarkers [8,10]. Differently from what was found in this study, in the only validation study comparing an inflammatory index of the diet, based on food parameters such as nutrients and phytochemicals, against hs-CRP in children from a North American population, a significant correlation was found between hs-CRP and the index in the highest quartiles [8]. It is noted in this study that the index was calculated using micronutrients, phytochemicals, and macronutrients; it is possible that for this reason, the results differed from our study. There are other studies evaluating the association of food groups with hs-CRP, where high hs-CRP levels were associated with a higher consumption of sweets and processed foods and with a low consumption of fruits and vegetables [4,5], which also differs from the findings of this study.

For adolescents, the findings of this study are in line with studies carried out in young European populations; these studies showed weak correlations between hs-CRP (and other biomarkers) with an inflammatory dietary score. In a study of Portuguese adolescents assessing the association of an inflammatory index of the diet with some inflammatory biomarkers, including hs-CRP, what was found was an association with other biomarkers (IL-6 and C4), but not with hs-CRP. An interesting finding of this study was that the index was associated with the general inflammatory biomarker [16], as shown in review studies with adults [7]. In another study of European adolescents, an inflammatory index of the diet was associated with other biomarkers, such as TNF-α, IL-1, 2, IFN-γ, and adhesion molecules. However, they were not associated with hs-CRP or IL-6 [30].

### 4.5. Limitations

This study has some limitations. First, there was a loss in the sample for reliability analyses (loss in the second measurement of the FFQ) and also in the validity analyses (failure to complete the 24HDR), which may have been caused by the extensive list that the questionnaires present, causing fatigue in filling out the questionnaire [42]. This presents a challenge in conducting large studies in the pediatric population using subjective tools for measuring food consumption. Despite that, our sample was large enough for this type of analysis according to the literature [22,23]. For financial reasons, our sample for criterion validity (FFQ1 × hs-CRP) was reduced. Although the post hoc analysis was shown to be positive, this fact may have influenced the results of the correlation coefficients.

Additionally, using only one inflammatory biomarker for criterion validity is a limitation. Studies with other populations show better correlation coefficients when associated with inflammatory indices with more than one inflammatory biomarker; however, there is no consensus in the literature on which biomarkers should be used [16,30].

### 4.6. Strengths

This study had some strengths. This is the first epidemiological study to validate a tool that measures the inflammatory potential of the diet in children and adolescents in South America. Our innovation was to develop and test the reliability and validity of the scores in multiethnic populations from six countries in South America, which, to date, does not exist in the literature. The study included a comparison of subjective (PAIFIS) and objective (biomarker) measures in a young and multicultural population from low- and middle-income countries, an area of research that was previously limited to high-income countries [8,16]; therefore, this study adds significantly to the body of literature in this area of research. In addition to usual individual adjustments such as gender, age, energy, and multilevel analysis, adjustments were made for contextual factors, including the city and school level, which was not undertaken in similar studies, which were limited to usual individual adjustments such as gender, age, and energy.

Finally, our score (PAIFIS) was calculated using food instead of macronutrients, micronutrients, and food components like other indices, thus following the trend of the new dietary guidelines for countries such as Brazil, Uruguay, Chile, and other South American countries. According to the new classification, foods are categorized by their level of processing [46], highlighting the importance of focusing on food and not on micro- and macronutrients because we eat foods and not just components. In addition, using foods, not food components, both in clinical practice and for epidemiological research, is a more practical way and can also minimize possible errors in measurements of diet components. In a Brazilian study of an adult population of females, which already used this new food classification, ultra-processed foods were associated with higher levels of hs-CRP, showing a possible path for future research in the pediatric population using this classification of food [47].

## 5. Conclusions

The PAIFIS proved to be a reliable tool for the pediatric population to estimate an inflammatory score based on an FFQ. For children and adolescents, it presented weak convergent validity and weak criterion validity. It is noted that in both age groups, the consumption of foods that compose the PAIFIS has a strong cultural influence, as evidenced in the multilevel regression analysis. More studies need to be carried out in this population, including other biomarkers, such as Il-6, TNF-alpha, adiponectin, and interleukins.

## Figures and Tables

**Figure 1 ijerph-20-01038-f001:**
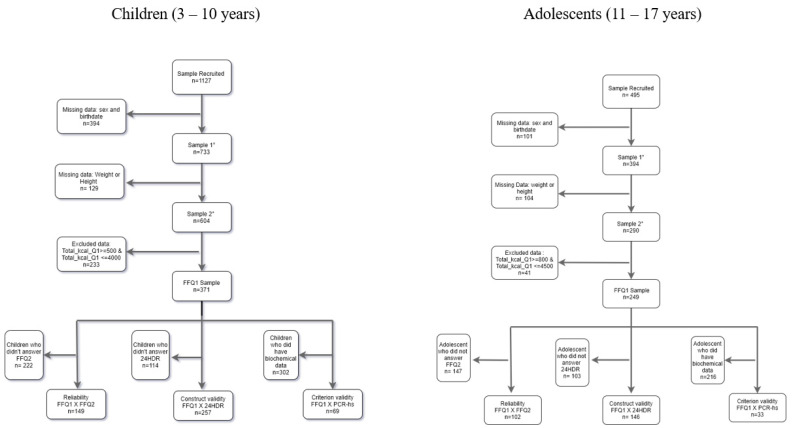
Flowchart that indicates the sampling process according to the type of analysis to be performed.

**Table 1 ijerph-20-01038-t001:** Composition of samples with the number of participants by center (SAYCARE) for reliability and validity of PAIFIS using an FFQ for inflammatory foods.

	São Paulo	Teresina	Buenos Aires	Medellin	Lima	Montevideo	Santiago	Total
Children								
Reliability	10	18	11	49	13	22	26	149
Convergent Validity	45	26	5	72	49	24	36	257
Criterion Validity	47	8	6	0	8	0	0	69
Adolescents								
Reliability	12	7	0	20	39	5	19	102
Convergent Validity	43	8	0	20	40	4	31	146
Criterion Validity	15	4	2	0	12	0	0	33

Pro-inflammatory/Anti-inflammatory Food Intake Score (PAIFIS).

**Table 2 ijerph-20-01038-t002:** Characteristics of children and adolescents from a sub-sample for reliability and validity analyses of the SAYCARE Study.

	Reliability				Criterion Validity				Criterion Validity			
Independent variables	Children (n = 149)		Adolescents (n = 102)		Children (n = 257)		Adolescents (n = 146)		Children (n = 69)		Adolescents (n = 33)	
	% or Median	(IQR)	% or Median	(IQR)	% orMedian	(IQR)	% or Median	(IQR)	% or Median	(IQR)	% or Median	(IQR)
Age	6.2	(4.5 9.1)	14.5	(12.4; 16.5)	6.9	(4.7; 8.8)	14.5	(14.1; 14.9)	7.8	(6.9; 9.6)	14.3	(12.4; 16.6)
Biological Sex												
Female	54.4	(46.3; 62.2)	55.8	(46.0; 65.3)	56.4	(50.3; 62.4)	50.7	(42.5; 58.8)	47.8	(36.2; 59.7)	48.5	(31.6; 65.7)
Male	45.3	(37.8; 53.7)	44.1	(34.7; 54.0)	43.6	(37.6; 49.7)	49.3	(41.2; 57.4)	52.2	(40.3; 63.8)	51.5	(34.3; 68.4)
Type of School												
Public	58.4	(50.3; 66.1)	31.4	(23.0; 41.1)	51.4	(45.2; 57.5)	43.8	(36.0; 52.0)	46.4	(34.8; 58.3)	27.3	(23.0; 41.1)
Private	41.6	(33.9; 49.7)	68.2	(58.8; 77.0)	48.7	(42.5; 54.8)	56.1	(48.0; 64.1)	52.1	(41.7; 65.2)	72.7	(54.6; 85.5)
Nutritional Status												
Thinness	6.7	(3.6; 12.1)	3.9	(1.5; 10.1)	5.8	(3.5; 9.5)	2.7	(1.0; 7.1)	2.9	(0.4; 19.3)		
Normal Weight	67.1	(59.1; 74.2)	62.7	(52.9; 71.7)	70.4	(64.5; 75.7)	65.1	(56.9; 72.4)	73.6	(55.7; 86.0)	66.7	(48.5; 80.9)
Overweight	17.4	(12.1; 24.5)	26.5	(18.7; 36.0)	13.2	(9.6; 18.0)	24	(17.7; 31.6)	17.6	(7.9; 34.9)	33.3	(19.1; 51.5)
Obesity	8.7	(5.11; 14.5)	6.8	(3.3; 13.8)	10.5	(7.3; 14.9)	8.2	(4.7; 14.0)	5.9	(1.4; 21.6)		
Waist circumference (cm)	54.5	(51.3; 61.2)	72.4	(67.9; 78.4)	55.9	(52.0; 62.0)	74.3	(72.7; 75.9)	57.8	(53.2; 63.5.1)	72.0	(67.0; 77.3)
hs-CRP (mg/dl)									0.3	(0.2; 0.72)	0.32	(0.09; 0.8)

Interquartile Range (IQR); body mass index (BMI); waist circumference (WC); high-sensitivity C-reactive protein (CRP).

**Table 3 ijerph-20-01038-t003:** Reliability of the PAIFIS in grams per day in a sub-sample of the SAYCARE Study.

	Children						Adolescents					
		FFQ1		FFQ2				FFQ1		FFQ2		
	n	Median	(IQR)	Median	(IQR)	ρ	n	Median	(IQR)	Median	(IQR)	ρ
Pro-inflammatory food group												
Sugar-sweetened Beverages	146	150.6	(49.9; 335.1)	149.3	(47.9; 309.8)	**0.66**	97	155.0	(15.0; 377.5)	218.5	(104.2; 439.9)	**0.59**
Processed meat	146	17.6	(7.4; 39.8)	16.4	(8.2; 33.9)	**0.63**	93	17.6	(0.3; 48.5)	23.4	(12.6; 48.5)	**0.54**
Red meat	141	43.1	(21.0; 86.0)	43.2	(21.5; 86.0)	**0.54**	93	28.0	(0.5; 86.1)	50.0	(17.5; 68.4)	**0.57**
Candies	149	122.9	(146.4; 205.4)	113.7	(47.3; 203.45)	**0.51**	102	90.6	(20.8 252.1)	125.6	(50.7; 251.3)	**0.31**
Snacks	136	22.4	(9.1; 47.5)	21.0	(7.7; 40.9)	**0.56**	96	21.1	(2.2; 47.6)	28.0	(14.0; 51.2)	**0.42**
Anti-inflammatory food group												
Fruits	149	231.8	(101.6; 420.0)	176.1	(9.0.8; 320,)	**0.58**	97	137.6	(25.9; 325.2)	206.0	(94.4; 370.0)	**0.53**
Vegetables	133	32.5	(14.0; 71.6)	28.5	(12.5; 61.0)	**0.59**	75	35.9	(10.8; 79.0)	30.1	(10.8; 79.0)	**0.47**
Sum groups												
Pro-inflammatory	149	440.5	(298.9; 765.9)	441.5	(244.5; 676.9)	**0.57**	102	491.3	(130.8 880.5)	501.2	(303.9; 930.3)	**0.52**
Anti-inflammatory	149	283.1	(124.8; 480.1)	212.5	(122.6; 380.0)	**0.63**	97	304.4	(174.5; 406.8)	238.1	(126.6; 420.3)	**0.55**
PAIFIS = Σ AI − Σ PI	149	222.5	(−13.24; 483.8)	155.4	(31.8; 462.1)	**0.69**	102	151.9	(1.7; 584.5)	266.5	(43.4; 635.5)	**0.61**

Interquartile Range (IQR); food frequency questionnaire (FFQ); 24 h Food Recall (24HDR); Pro-inflammatory/Anti-inflammatory Food Intake Score (PAIFIS); PAIFIS = Σ AI − Σ PI; ρ, Spearman’s rho coefficient. Spearman’s efficiency coefficients in bold show moderate or strong values.

**Table 4 ijerph-20-01038-t004:** Convergent validity of the PAIFIS in grams per day in a sub-sample of the SAYCARE Study.

**Children**		**FFQ1**		**R24hrs**				
	**n**	**Median**	**(IQR)**	**Median**	**(IQR)**	**ρ**	** *R ** **	**PCV ***
Pro-inflammatory food group								
Sugar-sweetened beverages	224	114.05	(21.8; 267.5)	360.0	(133.3; 653.3)	0.22	-	-
Processed meat	257	11.3	(0.3; 25.7)	10.1	(0.13; 36.8)	**0.37**	-	-
Red meat	257	25.2	(7.1; 57.0)	50,1	(1.1; 86.7)	**0.31**	-	-
Candies	257	85.5	(25.9 200.1)	83.8	(34.3; 149.3)	**0.4**	-	-
Snacks	257	16.1	(2.2; 31.8)	1.2	(0.1; 53.4)	0.28	-	-
Anti-inflammatory food group								
Fruits	232	172.2	(58.6; 373,7)	128.3	(56.7; 250.0)	0.22	-	-
Vegetables	228	26.9	(10.8; 68.4)	70.0	(30.0; 148.7)	0.23	-	-
Sum groups							-	-
Pro-inflammatory	257	369.8	(149.2;663.4)	559.0	(327.8; 934.5)	**0.36**	-	-
Anti-inflammatory	257	204.6	(77.6; 446.6)	215.1	(125.2; 361.67)	0.20	-	-
PAIFIS	257	72.4	(−22.93; 374.9)	350.4	(55.9; 695.9)	0.23	0.32	43.34
**Adolescents**		**FFQ1**		**R24hrs**				
	**n**	**Median**	**(IR)**	**Median**	**(IR)**	**ρ**	** *R ** **	**PCV ***
Pro-inflammatory food group								
Sugar-sweetened beverages	133	200.1	(50.6; 42.6)	520.0	(200.0; 903.3)	0.15	-	-
Processed meat	95	21.2	(0,5; 46.9)	51.18	(41.51; 60.86)	0.09	-	-
Red meat	108	42.1	(10.5; 93.0)	74.22	(64.40; 84.05)	0.24	-	-
Candies	136	181.8	(47.7; 273.6)	60.0	(26.3; 111.7)	0.18	-	-
Snacks	81	27.2	(.9.2; 53.9)	33.3	(0; 66.7)	0.25	-	-
Anti-inflammatory food group								
Fruits	110	180.6	(76.8; 386.0)	66.6	(0; 160.0)	0.16	-	-
Vegetables	121	35.9	(10.8; 79.0)	76.7	(32.5; 133.3)	0.17	-	-
Sum groups							-	-
Pro-inflammatory	146	540.5	(274.2; 964.3)	740.8	(386.7; 1157.9)	0.26	-	-
Anti-inflammatory	146	213.0	(104.1; 466.7)	163.3	(80.0; 263.33)	0.29	-	-
PAIFIS	146	247.8	(1.7; 640.7)	563.7	(207.1; 992.7)	0.16	0.24	15.45

Interquartile Range (IQR); food frequency questionnaire (FFQ); 24 h Food Recall (24HDR); Pro-inflammatory/Anti-inflammatory Food Intake Score (PAIFIS); PAIFIS = Σ AI − Σ PI. ρ, Spearman’s rho coefficient; PCV, proportional change attributable to region-level variance; R, regression coefficient; PAIFIS = Σ AI − Σ PI (g/day). * Coefficient from multilevel regression models after adjusting for center, type of school, sex, age, and total energy intake. Spearman’s efficiency coefficients in bold show moderate or strong values.

**Table 5 ijerph-20-01038-t005:** Validity criterion of the PAIFIS in grams per day with hs-CRP in a sub-sample of the SAYCARE Study.

		ΣPI (g/Day)	(ΣAI) (g/Day)	PAIFIS (g/Day)
Children (n = 69)	Median	407.08	214.70	102.2
	(IQR)	(134.8; 550.2)	(15.4; 417.8)	(1.7; 343.2)
	ρ	0.01	0.03	−0.03
	*R **	-	-	0.48
	PCV *	-	-	43.94
Adolescents (n = 33)	Median	531.7	205.7	225.7
	(IQR)	(259.1; 946.6)	(94.5 437.1)	(1.8; 640.7)
	ρ	0.16	0.07	0.24
	*R **	-	-	0.33
	PCV *	-	-	61.70

IQR, Interquartile Range; ρ, Spearman’s rho coefficient; PCV, proportional change attributable to region-level variance; R, regression coefficient; (ΣPI), sum for pro-inflammatory food group; ΣAI, sum for anti-inflammatory food group; PAIFIS, Σ AI − Σ PI. * Coefficient from multilevel regression models after adjusting for center, type of school, sex, age, and total energy intake.

## Data Availability

The datasets supporting the conclusions of this article can be accessed by reasonable request to the corresponding author: Augusto César F. De Moraes.

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
