# Peer review of "Reliability and Validity Estimate of the Pro-Inflammatory/Anti-Inflammatory Food Intake Score in South American Pediatric Population: SAYCARE Study"

_ijerph, 2023, doi:10.3390/ijerph20021038_

Round 1
Reviewer 1 Report
Dear authors,
Thank you for your manuscript. It is generally well written. The language could be improved in some places.
However, I still have a few aspects:
1.) Is no distinction made between AI and PI scores for food? I can hardly imagine that every type of fruit is included in exactly the same way. The sugar content alone between pineapple, figs and dates compared to strawberries or watermelon is clearly different. Please add another detailed paragraph regarding classification and calculation.
2.)They give the total scores and the respective gram numbers in their results section. However, the body weight also plays an important role. Please calculate them also on g body weight or ideally even on the FFM.
3.) In the discussion, we will only briefly look at existing data from Europe. Please compare your results more closely with such existing preliminary work. Possibly also in relation to body weight.
Author Response
1.) Is no distinction made between AI and PI scores for food? I can hardly imagine that every type of fruit is included in exactly the same way. The sugar content alone between pineapple, figs and dates compared to strawberries or watermelon is clearly different. Please add another detailed paragraph regarding classification and calculation.
Answer: Fruits were included in the AI since they are fresh and unprocessed fruits, as is well described in the literature, all fresh fruits contain micronutrients which in turn have an anti-inflammatory function, so fruits due to these characteristics were obviously not included in the score. pro inflammatory.
We did not make this comparison of fructose levels between fruits since our objective was to estimate the consumption of food items.
2.)They give the total scores and the respective gram numbers in their results section. However, the body weight also plays an important role. Please calculate them also on g body weight or ideally even on the FFM.
Answer :Our objective was not to calculate adequate consumption according to body weight, what we are validating is the consumption score of PI and AI foods that are independent of body weight, since this estimate is valid for the calculation of total calorie consumption and macronutrients.
3.) In the discussion, we will only briefly look at existing data from Europe. Please compare your results more closely with such existing preliminary work. Possibly also in relation to body weight.
Answer :Regarding data from Europe, validation of similar scores was only carried out in Europe and North America. Recently, studies were published with pediatric populations in South America, however, which tested associations with atherogenic risk (doi: 10.1017/S1368980021001816) and skipping breakfast (doi: 10.1016/j.nut.2022.111749)
Our innovation was to develop and test the reliability and validity of the scores in multiethnic populations from six countries in South America, which until now do not exist in the literature.
AS HIGHLIGHTED BELOW
"This is the first epidemiological study to validate a tool that measures the inflammatory potential of the diet in children and adolescents in South America. Our innovation was to develop and test the reliability and validity of the scores in multiethnic populations from six countries in South America, which until now do not exist in the literature. The study included a comparison of a subjective (PAIFIS) and objective (biomarker) measure in a young and multicultural population from low- and middle-income countries, an area of ​​research that was previously limited to high-income countries (8, 16); therefore, this study adds significantly to the body of literature in this area of ​​research. In addition to usual individual adjustments such as gender, age, energy, the multilevel analysis, adjustments were made for contextual factors, including city and school level, which was not done in similar studies, in addition to usual individual adjustments such as gender, age, energy."
Reviewer 2 Report
ijerph-2015306
Reliability and Validity Estimate of the Pro-inflammatory/Anti-inflammatory food intake score South American pediatric population: SAYCARE Study
Dear authors,
I have read your work with interest, although there are aspects that are not clear to me. In particular, the fact of using spearman correlation (for non-normal variables) but describing continuous variables with mean (suggesting that they are normal), creates confusion for me. I think you should review this. Another major issue is the lack of description of the PAIFIS score. You can find the rest of the comments below:
Abstract:
- Describe the abbreviations: FFQ, 24HDR, CRP
Introduction
- This sentence breaks the flow of the introduction. I suggest you use it in methodology, in the description of this variable.: “The literature suggests that the most commonly used inflammatory biomarker is hs-CRP. (7, 8)”
- Link the two paragraphs on dietary indices and describe abbreviations E-DII and C-DII
- Last line of the introduction: Use FFQ.
Methods
- Study design: Add information about the date on which the study was conducted and from which university or research centre the study is carried out. Also add information on ethical aspects (informed consent and ethics committees).
- Quality control of SACYARE Study: which software? Please, specify.
- Study Population and Data Collection for this Study: Did you use any software to calculate the sample size? If yes, please add this information.
- Food Frequency Questionnaire (FFQ): Are food frequency questionnaires the same for children and adolescents? Please clarify.
- 2.8. Pro-inflammatory/Anti-inflammatory food intake score (PAIFIS): Add more information on this score. It is not clear if it has been created by you or if it has been used previously in the literature. Add some reference for it.
- 2.9. Statistical analysis: Categorical variables are described with n and %, continuous variables with mean and SD if they have a normal distribution and with median and interquartile range if the distribution is not normal. Review the information you show on descriptive analysis.
Results
- Table 2: It would be useful to display categorical variables with n and % and continuous variables with mean and SD, in case they all follow a normal distribution.
- Table 3: I understand that continuous variables follow a normal distribution, since you show the mean, but you use spearman correlation which is for variables that do not follow a normal distribution. Please revise.
Discussion
- There is very little discussion. It would be advisable to further compare the results of the present study with those of other published studies, explaining what might explain the differences between the results of the present study and those of other published studies.
- Limitations: Were there differences between the characteristics of participants who dropped out and those who continued?
- Limitation: You should include recall bias and the fact that you have used the same core items in the FFQs for children and adolescents, when the diet between the two is often quite different.
Thank you
Author Response
Abstract:
- Describe the abbreviations: FFQ, 24HDR, CRP
Answer::Ok
Introduction
- This sentence breaks the flow of the introduction. I suggest you use it in methodology, in the description of this variable.: “The literature suggests that the most commonly used inflammatory biomarker is hs-CRP. (7, 8)”
Answer: Ok.
- Link the two paragraphs on dietary indices and describe abbreviations E-DII and C-DII
Answer: Ok
- Last line of the introduction: Use FFQ.
Answer: Ok
Methods
- Study design: Add information about the date on which the study was conducted and from which university or research centre the study is carried out. Also add information on ethical aspects (informed consent and ethics committees).
- Quality control of SACYARE Study: which software? Please, specify.
Answer:
Quality control of SACYARE Study
This multicenter feasibility study has the premise of ensuring that data collection is carried out in a standardized manner, achieving a good representation of reality and allowing the comparison of data from the different cities involved in the study. The data management system was created and led by a general coordinator of the study at the Medical School of the University of São Paulo. Reports from all centers were sent to São Paulo to verify the consistency of the data. For exclusively nutrition data: The questionnaire data (FFQ and R24HR) were entered into the software and tabulated by two independently trained nutritionists. Suppose there is any inconsistency, a data manager checks and corrects the error (20).
Also, to ensure the quality of the data, standardized procedures were applied, guaranteeing the accuracy and consistency of the results obtained during the study, such as manuals or standardized measurements, registration protocols and training of the entire team, and supervision of researchers through audit protocols. Inconsistencies were quickly identified and corrected with this procedure, diagnoses and resolution of difficulties were anticipated during data collection and processing.
Ethical aspects are located at the end of the document
- Study Population and Data Collection for this Study: Did you use any software to calculate the sample size? If yes, please add this information.
Answer:The sample and all statistical analyzes were performed using the Stata software.
- Food Frequency Questionnaire (FFQ): Are food frequency questionnaires the same for children and adolescents? Please clarify.
Answer: The only item that was included in the FFQ of the adolescents was alcohol and remembering that those who answered the children's questionnaire were the parents or guardians, as adolescents it was answered by themselves.
- 2.8. Pro-inflammatory/Anti-inflammatory food intake score (PAIFIS): Add more information on this score. It is not clear if it has been created by you or if it has been used previously in the literature. Add some reference for it.
Answer: Due to the wide food culture among the countries participating in the study, we decided to develop these specific scores for pediatric populations in South America instead of adapting existing ones in the literature. For this reason, we emphasize that this is an original study.
- 2.9. Statistical analysis: Categorical variables are described with n and %, continuous variables with mean and SD if they have a normal distribution and with median and interquartile range if the distribution is not normal. Review the information you show on descriptive analysis.
Answer: The variables have a normal distribution so we present them in means, however instead of presenting SD we present the CI since our study is epidemiological and for this characteristic it is important to present this population indicator.
Results
- Table 2: It would be useful to display categorical variables with n and % and continuous variables with mean and SD, in case they all follow a normal distribution.
Answer: The variables have a normal distribution so we present them in means, however instead of presenting SD we present the CI since our study is epidemiological and for this characteristic it is important to present this population indicator.
- Table 3: I understand that continuous variables follow a normal distribution, since you show the mean, but you use spearman correlation which is for variables that do not follow a normal distribution. Please revise.
Answer: You are correct, these variables in Table 3 do not have a normal distribution, for this reason we used Spearman's correlation coefficient. As previously stated, we prefer to present the mean and CI rather than the median and interquartile range for the main reason that the study is epidemiological and the CI is much more informative than the interquartile range.
Discussion
- There is very little discussion. It would be advisable to further compare the results of the present study with those of other published studies, explaining what might explain the differences between the results of the present study and those of other published studies
Answer: Regarding data from Europe, validation of similar scores was only carried out in Europe and North America. Recently, studies were published with pediatric populations in South America, however, which tested associations with atherogenic risk (doi: 10.1017/S1368980021001816) and skipping breakfast (doi: 10.1016/j.nut.2022.111749)
Our innovation was to develop and test the reliability and validity of the scores in multiethnic populations from six countries in South America, which until now do not exist in the literature.
Therefore, in order not to imply a vague discussion error by referencing association studies, which is not our objective, we chose to refer to validation studies in adults and children in other countries for this reason the brief discussion.
- Limitations: Were there differences between the characteristics of participants who dropped out and those who continued?
Answer: As highlighted by Heraclito et al. 2018 https://doi.org/10.1002/oby.22117 () we did not find differences for the main variables that we evaluated (SEX, age and mother's education in both age groups), however we found that children and adolescents from public schools had more losses in the second questionnaire.
An important result is that the highest proportions of losses/refusal were from individuals with lower SES, indicating that future studies in South America should oversample this SES class. This difficulty is attributed to the fact that because the questionnaire is extensive, the parents of the children and adolescents may have lost the motivation to respond to the same questionnaire twice in a short time period.
However, our n was within the estimated range.
- Limitation:
You should include recall bias
Answer: Ok
and the fact that you have used the same core items in the FFQs for children and adolescents, when the diet between the two is often quite different.
Answer: In epidemiological studies of this magnitude (More than 400 children in 7 cities and 6 countries) this bias is not differential since it is dispersed in the sample.
Reviewer 3 Report
If multicenter epidemiological studies suggest that a diet rich in raw vegetables, olive oil, fish, whole grains, and fruits is associated with low levels of inflammation why this others food groups were not considered in Anti-inflammatory profile analyses?
Could not find Supplementary Table 1.
Author Response
If multicenter epidemiological studies suggest that a diet rich in raw vegetables, olive oil, fish, whole grains, and fruits is associated with low levels of inflammation why this others food groups were not considered in Anti-inflammatory profile analyses?
ANSWER: Vegetables and fruits were considered, however olive, whole grains and fish had negligible consumption so we decided not to include them in order not to exceed zeros in the calculation formula.
Round 2
Reviewer 1 Report
Dear authors,
unfortunately, you have not responded adequately to any of my comments. Even though it is a validation study, you can discuss the results in more detail with existing observations.
In addition, you can include the factor with the different foods (e.g. fruits) and their individual consideration in the case in the limitations. The usefulness of a general index should also be critically questioned, whether it does not make more sense to calculate it in relation to body weight.
Author Response
Revisor 1
Comments and Suggestions for Authors
Dear authors,
unfortunately, you have not responded adequately to any of my comments. Even though it is a validation study, you can discuss the results in more detail with existing observations.
In addition, you can include the factor with the different foods (e.g. fruits) and their individual consideration in the case in the limitations. The usefulness of a general index should also be critically questioned, whether it does not make more sense to calculate it in relation to body weight.
Response: Dear Reviewer #1
We are so sorry if our responses do not meet your expectations. We've reviewed and adjusted it for your understanding, including suggested discussions and limitations. However, we want to emphasize that our main objective is to validate an instrument and not to make associations; because this should only be done after the instrument has been validated and with other samples and populations.
And this does not invalidate our study, since your suggestion to adjust the calculations for body weight (which is not our proposal) can be done in association studies.
We hope that in this new version, we have answered your criticisms satisfactorily.
Thank you
Reviewer 2 Report
Dear authors,
Thank you for your responses to my comments. It would be important that the explanations and clarifications you have given me are included in the article. For example, the main difference between the CFA of adolescents and children is the inclusion of the alcohol item.
Despite your answers, there are still some aspects to be clarified.
Abstract:
- - Please, put the abbreviations (FFQ, 24HDR, CRP) in the same way as you did with “PAIFIS”.
Methods
- - 2.3. Study Population and Data Collection for this Study: you indicated 3-17 years of age instead of 3-18.
- - Descriptive analysis: Thank you for your answer. First, the Shapiro-Wilk test is used in databases with up to 50 participants, so you should have used the Kolmogorov-Smirnov test to check the normality of the quantitative variables. Second, in the article you have written that categorical variables will be represented with % and CI. This is totally incorrect, it should be with n and %. I am familiar with epidemiological studies and understand that quantitative variables can be represented with mean and SD or median and RI, without much impact, as long as you take this into account in bivariate analyses. However, it is not entirely correct to show mean and SD when the quantitative variable is non-parametric.
Results
- -Table 1: add %
- - Table 2: with the new information is very difficult to understand.
- - Table 4: Possible grammatical mistake in PAIFIS
Discussion
- - Thank you for your response to my comments on discussion. However, I continue to think that the discussion could be enriched a little more, as you have simply summarised your results in a narrative form. Although there are no validation studies in your population, you can compare your findings with those found in other validation studies in children and adolescents, which do exist.
Limitations
- - Add the information and reference with which you have responded to my previous comment regarding the difference between the participants who took part and those who dropped out.
Thank you
Author Response
Revisor 2
Dear authors,
Thank you for your responses to my comments. It would be important that the explanations and clarifications you have given me are included in the article. For example, the main difference between the CFA of adolescents and children is the inclusion of the alcohol item.
Response: Thank you we included that on the new version.
Despite your answers, there are still some aspects to be clarified.
Abstract:
- - Please, put the abbreviations (FFQ, 24HDR, CRP) in the same way as you did with “PAIFIS”.
Response: Thank you we included that on the new version.
Methods
- - 2.3. Study Population and Data Collection for this Study: you indicated 3-17 years of age instead of 3-18.
Response: Thank you we included that on the new version.
- - Descriptive analysis: Thank you for your answer. First, the Shapiro-Wilk test is used in databases with up to 50 participants, so you should have used the Kolmogorov-Smirnov test to check the normality of the quantitative variables. Second, in the article you have written that categorical variables will be represented with % and CI. This is totally incorrect, it should be with n and %. I am familiar with epidemiological studies and understand that quantitative variables can be represented with mean and SD or median and RI, without much impact, as long as you take this into account in bivariate analyses. However, it is not entirely correct to show mean and SD when the quantitative variable is non-parametric.
Response: Thank you for the suggestion, we included the median and interquartile interval.
Results
- -Table 1: add %
Response: Thank you we included that on the new version.
- - Table 2: with the new information is very difficult to understand.
- - Table 4: Possible grammatical mistake in PAIFIS
Response: Thank you we included that on the new version.
Discussion
- - Thank you for your response to my comments on discussion. However, I continue to think that the discussion could be enriched a little more, as you have simply summarised your results in a narrative form. Although there are no validation studies in your population, you can compare your findings with those found in other validation studies in children and adolescents, which do exist.
Response: Thank you for the suggestion, we included a sentence about validation studies in pediatric population. Highlighted bellow:
“When studying relationships between dietary intake and disease development in epidemiology, it is important to account for the dietary assessment method’s ability to categorize participants by dietary intake correctly. Studies to assess the reliability of food frequency questionnaires in a free-living environment will be explored in comparison with instruments that provide objective measures to improve the accuracy of subjective measures and is common in nutritional epidemiology (22). The SAYCARE FFQ was validated for fruit and vegetable intake (24) and also for iron intake (47) with very good coefficients (range 0.35 to 0.85). Consequently, the FFQ should not be used as a quantitative tool to measure the anti-pro inflammatory status of pediatric populations.
Limitations
- - Add the information and reference with which you have responded to my previous comment regarding the difference between the participants who took part and those who dropped out.
Response: Thank you we included that on the new version.
Thank you